# Privacy-Preserving Classification of Personal Text Messages with Secure Multi-Party Computation: An Application to Hate-Speech Detection

**Devin Reich**[1], **Ariel Todoki**[1], **Rafael Dowsley**[2], **Martine De Cock**[1]*, **Anderson Nascimento**[1]
[1] School of Engineering and Technology
University of Washington Tacoma
Tacoma, WA 98402
{dreich,atodoki,mdecock,andclay}@uw.edu
[2]Department of Computer Science
Bar-Ilan University, 5290002, Ramat-Gan, Israel
rafael@dowsley.net

## Abstract

Classification of personal text messages has many useful applications in surveillance, e-commerce, and mental health care, to name a few. Giving applications access to personal texts can easily lead to (un)intentional privacy violations. We propose the first privacy-preserving solution for text classification that is provably secure. Our method, which is based on Secure Multiparty Computation (SMC), encompasses both feature extraction from texts, and subsequent classification with logistic regression and tree ensembles. We prove that when using our secure text classification method, the application does not learn anything about the text, and the author of the text does not learn anything about the text classification model used by the application beyond what is given by the classification result itself. We perform end-to-end experiments with an application for detecting hate speech against women and immigrants, demonstrating excellent runtime results without loss of accuracy.

## 1 Introduction

The ability to elicit information through automated scanning of personal texts has significant economic and societal value. Machine learning (ML) models for classification of text such as e-mails and SMS messages can be used to infer whether the author is depressed [46], suicidal [42], a terrorist threat [1], or whether the e-mail is a spam message [2, 49]. Other valuable applications of text message classification include user profiling for tailored advertising [32], detection of hate speech [6], and detection of cyberbullying [51]. Some of the above are integrated in parental control applications[2] that monitor text messages on the phones of children and alert their parents when content related to drug use, sexting, suicide etc. is detected. Regardless of the clear benefits, giving applications access to one's personal text messages and e-mails can easily lead to (un)intentional privacy violations.

In this paper, we propose the first privacy-preserving (PP) solution for text classification that is provably secure. To the best of our knowledge, there are no existing Differential Privacy (DP) or Secure Multiparty Computation (SMC) based solutions for PP feature extraction and classification of unstructured texts; the only existing method is based on Homomorphic Encryption (HE) and takes 19 minutes to classify a tweet [15] while leaking information about the text being classified. In our SMC

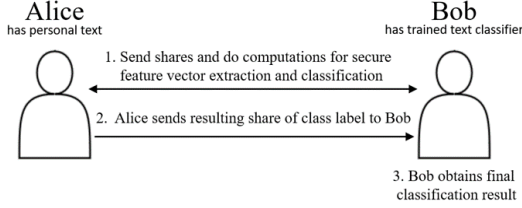

Figure 1: Roles of Alice and Bob in SMC based text classification

based solution, there are two parties, nick-named *Alice* and *Bob* (see Fig. 1). Bob has a trained ML model that can automatically classify texts. Our secure text classification protocol allows to classify a personal text written by Alice with Bob's ML model in such a way that Bob does not learn anything about Alice's text and Alice does not learn anything about Bob's model. Our solution relies on PP protocols for feature extraction from text and PP machine learning model scoring, which we propose in this paper.

We perform end-to-end experiments with an application for PP detection of hate speech against women and immigrants in text messages. In this use case, Bob has a trained logistic regression (LR) or AdaBoost model that flags hateful texts based on the occurrence of particular words. LR models on word $n$-grams have been observed to perform comparably to more complex CNN and LSTM model architectures for hate speech detection [35]. Using our protocols, Bob can label Alice's texts as hateful or not without learning which words occur in Alice's texts, and Alice does not learn which words are in Bob's hate speech lexicon, nor how these words are used in the classification process. Moreover, classification is done in seconds, which is two orders of magnitude better than the existing HE solution despite the fact we use over 20 times more features and do not leak any information about Alice's text to the model owner (Bob). The solution based on HE leaks which words in the text are present in Bob's lexicon [15].

We build our protocols using a privacy-preserving machine learning (PPML) framework based on SMC developed by us[3] . All the existing building blocks can be composed within themselves or with new protocols added to the framework. On top of existing building blocks, we also propose a novel protocol for binary classification over binary input features with an ensemble of decisions stumps. While some of our building blocks have been previously proposed, the main contribution of this work consists of the careful choice of the ML techniques, feature engineering and algorithmic and implementation optimizations to enable end-to-end practical PP text classification . Additionally, we provide security definitions and proofs for our proposed protocols.

## 2   Preliminaries

We consider *honest-but-curious adversaries*, as is common in SMC based PPML (see e.g. [19, 21]). An honest-but-curious adversary follows the instructions of the protocol, but tries to gather additional information. Secure protocols prevent the latter.

We perform SMC using additively secret shares to do computations modulo an integer $q$. A value $x$ is secret shared over $\mathbb{Z}_q = \{0, 1, \ldots, q-1\}$ between parties Alice and Bob by picking $x_A, x_B \in \mathbb{Z}_q$ uniformly at random subject to the constraint that $x = x_A + x_B \mod q$, and then revealing $x_A$ to Alice and $x_B$ to Bob. We denote this secret sharing by $[\![x]\!]_q$, which can be thought of as a shorthand for $(x_A, x_B)$. Secret-sharing based SMC works by first having the parties split their respective *inputs* in secret shares and send some of these shares to each other. Naturally, these inputs have to be mapped appropriately to $\mathbb{Z}_q$. Next, Alice and Bob represent the *function* they want to compute securely as a circuit consisting of addition and multiplication gates. Alice and Bob will perform secure additions and multiplications, gate by gate, over the shares until the desired outcome is obtained. The final result can be recovered by combining the final shares, and disclosed as intended, i.e. to one of the parties or to both. It is also possible to keep the final result distributed over shares.

In SMC based text classification, as illustrated in Fig. 1, Alice's *input* is a personal text $x$ and Bob's *input* is an ML model $\mathcal{M}$ for text classification. The function that they want to compute securely is

$f(x, \mathcal{M}) = \mathcal{M}(x)$, i.e. the class label of $x$ when classified by $\mathcal{M}$. To this end, Alice splits the text in secret shares while Bob splits the ML model in secret shares. Both parties engage in a protocol in which they send some of the input shares to each other, do local computations on the shares, and repeat this process in an iterative fashion over shares of intermediate results (Step 1). At the end of the joint computations, Alice sends her share of the computed class label to Bob (Step 2), who combines it with his share to learn the classification result (Step 3). As mentioned above, the protocol for Step 1 involves representing the function $f$ as a circuit of addition and multiplication gates.

Given two secret sharings $[\![x]\!]_q$ and $[\![y]\!]_q$, Alice and Bob can locally compute in a straightforward way a secret sharing $[\![z]\!]_q$ corresponding to $z = x + y$ or $z = x - y$ by simply adding/subtracting their local shares of $x$ and $y$ modulo $q$. Given a constant $c$, they can also easily locally compute a secret sharing $[\![z]\!]_q$ corresponding to $z = cx$ or $z = x + c$: in the former case Alice and Bob just multiply their local shares of $x$ by $c$; in the latter case Alice adds $c$ to her share of $x$ while Bob keeps his original share. These local operations will be denoted by $[\![z]\!]_q \leftarrow [\![x]\!]_q + [\![y]\!]_q$, $[\![z]\!]_q \leftarrow [\![x]\!]_q - [\![y]\!]_q$, $[\![z]\!]_q \leftarrow c[\![x]\!]_q$ and $[\![z]\!]_q \leftarrow [\![x]\!]_q + c$, respectively. To allow for efficient secure multiplication of values via operations on their secret shares (denoted by $[\![z]\!]_q \leftarrow [\![x]\!]_q[\![y]\!]_q$), we use a trusted initializer that pre-distributes correlated randomness to the parties participating in the protocol before the start of Step 1 in Fig. 1.[4] The initializer is not involved in any other part of the execution and does not learn any data from the parties. This can be straightforwardly extended to efficiently perform secure multiplication of secret shared matrices. The protocol for secure multiplication of secret shared matrices is denoted by $\pi_{\mathsf{DMM}}$ and for the special case of inner-product computation by $\pi_{\mathsf{IP}}$. Details about the (matrix) multiplication protocol can be found in [19]. We note that if a trusted initializer is not available or desired, Alice and Bob can engage in pre-computations to securely emulate the role of the trusted initializer, at the cost of introducing computational assumptions in the protocol [19].

## 3    Secure text classification

Our general protocol for PP text classification relies on several building blocks that are used together to accomplish Step 1 in Fig. 1: a secure equality test, a secure comparison test, private feature extraction, secure protocols for converting between secret sharing modulo 2 and modulo $q > 2$, and private classification protocols. Several of these building blocks have been proposed in the past. However, to the best of our knowledge, this is the very first time they are combined in order to achieve efficient text classification with provable security.

We assume that Alice has a personal text message, and that Bob has a LR or AdaBoost classifier that is trained on unigrams and bigrams as features. Alice constructs the set $A = \{a_1, a_2, \ldots, a_m\}$ of unigrams and bigrams occurring in her message, and Bob constructs the set $B = \{b_1, b_2, \ldots, b_n\}$ of unigrams and bigrams that occur as features in his ML model. We assume that all $a_j$ and $b_i$ are in the form of bit strings. To achieve this, Alice and Bob convert each unigram and bigram on their end to a number $N$ using SHA 224 [44], strictly for its ability to map the same inputs to the same outputs in a pseudo-random manner. Next Alice and Bob map each $N$ on their end to a number between 0 and $2^l - 1$, i.e. a bit string of length $l$, using a random function in the universal hash family proposed by Carter and Wegman [12].[5] In the remainder we use the term "word" to refer to a unigram or bigram, and we refer to the set $B = \{b_1, b_2, \ldots, b_n\}$ as Bob's lexicon.

Below we outline the protocols for PP text classification. A correctness and security analysis of the protocols is provided as an appendix. In the description of the protocols in this paper, we assume that Bob needs to learn the result of the classification, i.e. the class label, at the end of the computations. It is important to note that the protocols described below can be straightforwardly adjusted to a scenario where Alice instead of Bob has to learn the class label, or even to a scenario where neither Alice nor Bob should learn what the class label is and instead it should be revealed to a third party or kept in a secret sharing form. All these scenarios might be relevant use cases of PP text classification, depending on the specific application at hand.

### 3.1 Cryptographic building blocks

**Secure Equality Test:** At the start of the secure equality test protocol, Alice and Bob have secret shares of two bit strings $x = x_\ell \ldots x_1$ and $y = y_\ell \ldots y_1$ of length $\ell$. $x$ corresponds to a word from Alice's message and $y$ corresponds to a feature from Bob's model. The bit strings $x$ and $y$ are secret shared over $\mathbb{Z}_2$. Alice and Bob follow the protocol to determine whether $x = y$. The protocol $\pi_{\mathsf{EQ}}$ outputs a secret sharing of 1 if $x = y$ and of 0 otherwise.

Protocol $\pi_{\mathsf{EQ}}$:
- For $i = 1, \ldots, \ell$, Alice and Bob locally compute $[\![r_i]\!]_2 \leftarrow [\![x_i]\!]_2 + [\![y_i]\!]_2 + 1$.
- Alice and Bob use secure multiplication to compute a secret sharing of $z = r_1 \cdot r_2 \cdot \ldots \cdot r_\ell$. If $x = y$, then $r_i = 1$ for all bit positions $i$, hence $z = 1$; otherwise some $r_i = 0$ and therefore $z = 0$. The result is the secret sharing $[\![z]\!]_2$, which is the desired output of the protocol.

This protocol for equality test is folklore in the field of SMC. The $l - 1$ multiplications can be organized in as binary tree with the result of the multiplication at the root of the tree. In this way, the presented protocol has $\log(l)$ rounds. While there are equality test protocols that have a constant number of rounds, the constant is prohibitively large for the parameters used in our implementation.

**Secure Feature Vector Extraction:** At the start of the feature extraction protocol, Alice has a set $A = \{a_1, a_2, \ldots, a_m\}$ and Bob has a set $B = \{b_1, b_2, \ldots, b_n\}$. $A$ is a set of bit strings that represent Alice's text, and $B$ is a set of bit strings that represent Bob's lexicon. Bob would like to extract words from Alice's text that appear in his lexicon. At the end of the protocol, Alice and Bob have secret shares of a binary feature vector $x$ which represents what words in Bob's lexicon appear in Alice's text. The binary feature vector $x$ of length $n$ is defined as

$$x_i = \begin{cases} 1 & \text{if } b_i \in A \\ 0 & \text{otherwise} \end{cases} \tag{1}$$

Protocol $\pi_{\mathsf{FE}}$:
- Alice and Bob secret share each $a_j$ $(j = 1, \ldots, m)$ and each $b_i$ $(i = 1, \ldots, n)$ with each other.
- For $i = 1 \ldots n$: // Computation of secret shares of $x_i$ as defined in Equation (1).
    For $j = 1 \ldots m$:
        Alice and Bob run the secure equality test protocol $\pi_{\mathsf{EQ}}$ to compute secret shares

$$x_{ij} = 1 \text{ if } a_j = b_i; \quad x_{ij} = 0 \text{ otherwise} \tag{2}$$

    Alice and Bob locally compute the secret share $[\![x_i]\!]_2 \leftarrow \sum_{j=1}^{m} [\![x_{ij}]\!]_2$.

The secure feature vector extraction can be seen as a private set intersection where the intersection is not revealed but shared [13, 31]. Our solution $\pi_{\mathsf{FE}}$ is tailored to be used within our PPML framework (it uses only binary operations, it is secret sharing based, and is based on pre-distributed binary multiplications). In principle, other protocols could be used here. The efficiency of our protocol can be improved by using hashing techniques [45] at the cost of introducing a small probability of error. The improvements due to hashing are asymptotic and for the parameters used in our fastest running protocol these improvements were not noticeable. Thus, we restricted ourselves to the original protocol without hashing and without any probability of failure.

**Secure Comparison Test:** In our privacy-preserving AdaBoost classifier we will use a secure comparison protocol as a building block. At the start of the secure comparison test protocol, Alice and Bob have secret shares over $\mathbb{Z}_2$ of two bit strings $x = x_\ell \ldots x_1$ and $y = y_\ell \ldots y_1$ of length $\ell$. They run the secure comparison protocol $\pi_{\mathsf{DC}}$ of Garay et al. [34] with secret sharings over $\mathbb{Z}_2$ and obtain a secret sharing of 1 if $x \geq y$ and of 0 otherwise.

**Secure Conversion between $\mathbb{Z}_q$ and $\mathbb{Z}_2$:** Some of our building blocks perform computations using secret shares over $\mathbb{Z}_2$ (secure equality test, comparison and feature extraction), while the secure inner product works over $\mathbb{Z}_q$ for $q > 2$. In order to be able to integrate these building blocks we need:
- A secure bit-decomposition protocol for secure conversion from $\mathbb{Z}_q$ to $\mathbb{Z}_2$: Alice and Bob have as input a secret sharing $[\![x]\!]_q$ and without learning any information about $x$ they should obtain as output secret sharings $[\![x_i]\!]_2$, where $x_\ell \cdots x_1$ is the binary representation of $x$. We use the secure bit-decomposition protocol $\pi_{\mathsf{decomp}}$ from De Cock et al. [19].

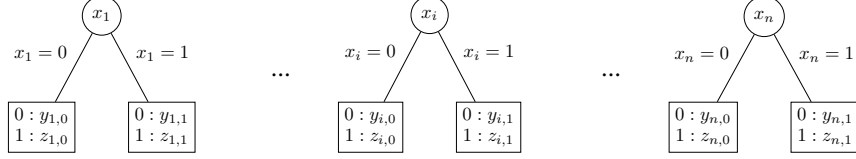

Figure 2: Ensemble of decision stumps. Each root corresponds to a feature $x_i$. The leaves contain weights $y_{i,k}$ for the votes for class label 0 and weights $z_{i,k}$ for the votes for class label 1.

- A protocol for secure conversion from $\mathbb{Z}_2$ to $\mathbb{Z}_q$: Alice and Bob have as a input a secret sharing $[\![x]\!]_2$ of a bit $x$ and need to obtain a secret sharing $[\![x]\!]_q$ of the binary value over a larger field $\mathbb{Z}_q$ without learning any information about $x$. To this end, we use protocol $\pi_{2\text{toQ}}$:
  - For the input $[\![x]\!]_2$, let $x_A \in \{0, 1\}$ denote Alice's share and $x_B \in \{0, 1\}$ denote Bob's share.
  - Alice creates a secret sharing $[\![x_A]\!]_q$ by picking uniformly random shares that sum to $x_A$ and delivers Bob's share to him, and Bob proceeds similarly to create $[\![x_B]\!]_q$.
  - Alice and Bob compute $[\![y]\!]_q \leftarrow [\![x_A]\!]_q [\![x_B]\!]_q$.
  - The output is computed as $[\![z]\!]_q \leftarrow [\![x_A]\!]_q + [\![x_B]\!]_q - 2[\![y]\!]_q$.

**Secure Logistic Regression (LR) Classification:**  At the start of the secure LR classification protocol, Bob has a trained LR model $\mathcal{M}$ that requires a feature vector $x$ of length $n$ as its input, and produces a label $\mathcal{M}(x)$ as its output. Alice and Bob have secret shares of the feature vector $x$ which represents what words in Bob's lexicon appear in Alice's text. At the end of the protocol, Bob gets the result of the classification $\mathcal{M}(x)$. We use an existing protocol $\pi_{\text{LR}}$ for secure classification with LR models [19].[6]

**Secure AdaBoost Classification:**  The setting is the same as above, but the model $\mathcal{M}$ is an AdaBoost ensemble of decision stumps instead of a LR model. While efficient solutions for secure classification with tree ensembles were previously known [33], we can take advantage of specific facts about our use case to obtain a more efficient solution. In more detail, in our use case: (1) all the decision trees have depth 1 (i.e., decision stumps); (2) each feature $x_i$ is binary and therefore when it is used in a decision node, the left and right children correspond exactly to $x_i = 0$ and $x_i = 1$; (3) the output class is binary; (4) the feature values were extracted in a PP way and are secret shared so that no party alone knows their values. We can use the above facts in order to perform the AdaBoost classification by computing two inner products and then comparing their values.

Protocol $\pi_{\text{AB}}$:
- Alice and Bob hold secret sharings $[\![x_i]\!]_q$ of each of the $n$ binary features $x_i$. Bob holds the trained AdaBoost model which consists of two weighted probability vectors $y = (y_{1,0}, y_{1,1}, \ldots, y_{n,0}, y_{n,1})$ and $z = (z_{1,0}, z_{1,1}, \ldots, z_{n,0}, z_{n,1})$. For the $i$-th decision stump: $y_{i,k}$ is the weighted probability (i.e., a probability multiplied by the weight of the $i$-th decision stump) that the model assigns to the output class being 0 if $x_i = k$, and $z_{i,k}$ is defined similarly for the output class 1 (see Fig. 2).
- Bob secret shares the elements of $y$ and $z$, and Alice and Bob locally compute secret sharings $[\![w]\!]_q$ of the vector $w = (1 - x_1, x_1, 1 - x_2, x_2, \ldots, 1 - x_n, x_n)$.
- Using the secure inner product protocol $\pi_{\text{IP}}$, Alice and Bob compute secret sharings of the inner product $p_0$ between $y$ and $w$, and of the inner product $p_1$ between $z$ and $w$. $p_0$ and $p_1$ are the aggregated votes for class label 0 and 1 respectively.
- Alice and Bob use $\pi_{\text{decomp}}$ to compute bitwise secret sharings of $p_0$ and $p_1$ over $\mathbb{Z}_2$.
- Alice and Bob use $\pi_{\text{DC}}$ to compare $p_1$ and $p_0$, getting as output a secret sharing of the output class $c$, which is then open towards Bob.

To the best of our knowledge, this is the most efficient provably secure protocol for binary classification over binary input features with an ensemble of decisions stumps.

Table 1: Accuracy (Acc) results using 5-fold cross-validation over the corpus of 10,000 tweets. Total time (Tot) needed to securely classify a text with our framework, broken down in time needed for feature vector extraction (Extr) and time for feature vector classification (Class).

| | Unigrams | | | | Unigrams+Bigrams | | | |
|---|---|---|---|---|---|---|---|---|
| | Acc | Time (in sec) | | | Acc | Time (in sec) | | |
| | | Extr | Class | Tot | | Extr | Class | Tot |
| Ada; 50 trees; depth 1 | 71.6% | 0.8 | 6.4 | 7.2 | 73.3% | 1.5 | 6.6 | 8.1 |
| Ada; 200 trees; depth 1 | 73.0% | 2.8 | 6.4 | 9.2 | 74.2% | 9.4 | 6.6 | 16.0 |
| Ada; 500 trees; depth 1 | 73.9% | 6.6 | 6.7 | 13.3 | **74.4%** | **21.6** | **6.7** | **28.3** |
| Logistic regression (50 feat.) | **72.4%** | **0.8** | **3.7** | **4.5** | 73.8% | 1.5 | 3.8 | 5.3 |
| Logistic regression (200 feat.) | 73.3% | 2.8 | 3.7 | 6.5 | 73.7% | 9.4 | 3.8 | 13.2 |
| Logistic regression (500 feat.) | 73.4% | 6.6 | 3.8 | 10.4 | 74.2% | 21.6 | 4.1 | 25.7 |
| Logistic regression (all feat.) | 73.1% | 318.0 | 6.1 | 324.1 | 73.8% | 5,371.9 | 24.9 | 5,396.8 |

## 3.2 Privacy-preserving classification of personal text messages

We now present our novel protocols for PP text classification. They result from combining the cryptographic building blocks we introduced previously. The PP protocol $\pi_{\mathsf{TC-LR}}$ for classifying the text using a logistic regression model works as follows:

Protocol $\pi_{\mathsf{TC-LR}}$:
- Alice and Bob execute the secure feature extraction protocol $\pi_{\mathsf{FE}}$ with input sets $A$ and $B$ in order to obtain the secret shares $[\![x_i]\!]_2$ of the feature vector $x$.
- They run the protocol $\pi_{\mathsf{2toQ}}$ to obtain shares $[\![x_i]\!]_q$ over $\mathbb{Z}_q$.
- Alice and Bob run the secure logistic regression classification protocol $\pi_{\mathsf{LR}}$ in order to get the result of the classification. The LR model $\mathcal{M}$ is given as input to $\pi_{\mathsf{LR}}$ by Bob, and the secret shared feature vector $x$ by both of them. Bob gets the result of the classification $\mathcal{M}(x)$.

The privacy-preserving protocol $\pi_{\mathsf{TC-AB}}$ for classifying the text using AdaBoost works as follows:

Protocol $\pi_{\mathsf{TC-AB}}$:
- Alice and Bob execute the secure feature extraction protocol $\pi_{\mathsf{FE}}$ with input sets $A$ and $B$ in order to obtain the secret shares $[\![x_i]\!]_2$ of the feature vector $x$.
- They run the protocol $\pi_{\mathsf{2toQ}}$ to obtain shares $[\![x_i]\!]_q$ over $\mathbb{Z}_q$.
- Alice and Bob run the secure AdaBoost classification protocol $\pi_{\mathsf{AB}}$ to obtain the result of the classification. The secret shared feature vector $x$ is given as input to $\pi_{\mathsf{AB}}$ by both of them, and the two weighted probability vectors $y = (y_{1,0}, y_{1,1}, \ldots, y_{n,0}, y_{n,1})$ and $z = (z_{1,0}, z_{1,1}, \ldots, z_{n,0}, z_{n,1})$ that constitute the model are specified by Bob. Bob gets the output class $c$.

Detailed proofs of security are presented in the appendix.

# 4 Experimental results

We evaluate the proposed protocols in a use case for the detection of hate speech in short text messages, using data from [6]. The corpus consists of 10,000 tweets, 60% of which are annotated as hate speech against women or immigrants. We convert all characters to lowercase, and turn each tweet into a set of word unigrams and bigrams. There are 29,853 distinct unigrams and 93,629 distinct bigrams in the dataset, making for a total of 123,482 features.

Accuracy results for a variety of models trained to classify a tweet as hate speech vs. non-hate speech are presented in Table 1. The models are evaluated using 5-fold cross-validation over the entire corpus of 10,000 tweets. The top rows in Table 1 correspond to tree ensemble models consisting of 50, 200, and 500 decision stumps respectively; the root of each stump corresponds to a feature. The bottom rows contain results for an LR model trained on 50, 200, and 500 features (preselected based on information gain), and an LR model trained on all features. We ran experiments for feature sets consisting of unigrams and bigrams, as well as for feature sets consisting of unigrams only, observing that the inclusion of bigrams leads to a small improvement in accuracy. Note that designing a model to obtain the highest possible accuracy is not the focus of this paper. Instead, our goal is to demonstrate that PP text classification based on SMC is feasible in practice.

We implemented the protocols from Section 3 in Java and ran experiments on AWS c5.9xlarge machines with 36 vCPUs, 72.0 GiB Memory.[7] Each of the parties ran on separate machines (connected with a Gigabit Ethernet network), which means that the results in Table 1 cover communication time in addition to computation time. Each runtime experiment was repeated 3 times and average results are reported. In Table 1 we report the time (in sec) needed for converting a tweet into a feature vector (Extr), for classification of the feature vector (Class), and for the overall process (Tot).

## 4.1 Analysis

The best running times were obtained using unigrams, 50 features and logistic regression (4.5 s) with an accuracy of 72.4%. The highest accuracy (74.4%) was obtained by using unigram and bigrams, 500 features and AdaBoost with a running time equal to 28.3s. From these results, it is clear that feature engineering plays a major role in optimizing privacy-preserving machine learning solutions based on SMC. We managed to reduce the running time from 5,396.8s (logistic regression, unigram and bigrams, all 123,482 features being used) to 5.3s (logistic regression, unigrams and bigrams, 50 features) without any loss in accuracy and to 4.5s (logistic regression, unigrams only, 50 features) with a small loss.

## 4.2 Optimizing the computational and communication complexities

The feature extraction protocol requires $n \cdot m$ secure equality tests of bit strings. The equality test relies on secure multiplication, which is the more expensive operation. To reduce the number of required equality tests, Alice and Bob can each first map their bit strings to $p$ buckets $A_1, A_2, \ldots, A_p$ and $B_1, B_2, \ldots, B_p$ respectively, so that bit strings from each $A_i$ need to only be compared with bit strings from $B_i$. Each bit string $a_j$ and $b_i$ is hashed and the first $t$ bits of the hash output are used to define the bucket number corresponding to that bit string, using a total of $p = 2^t$ buckets. In order not to leak how many elements are mapped to each bucket (which can leak some information about the probability distribution of the elements, as the hash function is known by everyone), each bucket has a fixed number of elements ($s_1$ for Bob's buckets and $s_2$ for Alice's buckets) and the empty spots in the buckets are filled up with dummy elements. The feature extraction protocol now requires $p \cdot s_1 \cdot s_2$ equality tests, which can be substantially smaller than $n \cdot m$. When using bucketization, the feature vector of length $n$ from Equation (1) is expanded to a feature vector of length $p \cdot s_1$, containing the original $n$ features as well as the $p \cdot s_1 - n$ dummy features that Bob created to fill up his buckets. These dummy features do not have any effect on the accuracy of the classification because Bob's model does not take them into account: the trees with dummy features in an AdaBoost model have 0 weight for both class labels, and the dummy features' coefficients in an LR model are always 0.

The size of the buckets has to be chosen sufficiently large to avoid overflow. The choice depends directly on the number $p = 2^t$ of buckets (which is kept constant for Alice and Bob) and the number of elements to be placed in the buckets, i.e. $n$ elements on Bob's side and $m$ elements on Alice's side. While for hash functions coming from a 2-universal family of hash functions the computation of these probabilities is relatively straightforward, the same is not true for more complicated hash functions [45]. In that case, numerical simulations are needed in order to bound the required probability.

The effect of using buckets is more significant for large values of $n$ and $m$. In our case, after performing feature engineering for reducing the number of elements in each set, in the best case, we end up with inputs for which there is no significant difference between the original protocol (without buckets) and the protocol that uses buckets. If the performance of these two cases is comparable, one is better off using the version without buckets, since there will be no probability of information being leaked due to bucket overflow.

Another way we could possibly improve the communication and computation complexities of the protocol is by reducing the number of bits used to represent each feature albeit at the cost of increasing the probability of collisions (different features being mapped into the same bit strings). We used 13 bits for representing unigrams and 17 bits for representing unigrams and bigrams. We did not observe any collisions.

Finally, we note that if the protocol is to be deployed over a wide area network, rather than a local area network, Yao garbled circuits would become a preferable choice for the round intensive parts of our solution (such as in the private feature extraction part).

# 5 Related work

The interest in privacy-preserving machine learning (PPML) has grown substantially over the last decade. The best-known results in PPML are based on differential privacy (DP), a technique that relies on adding noise to answers, to prevent an adversary from learning information about any particular individual in the dataset from revealed aggregate statistics [30]. While DP in an ML setting aims at protecting the privacy of individuals in the training dataset, our focus is on protecting the privacy of new user data that is classified with proprietary ML models. To this end, we use Secure Multiparty Computation (SMC) [16], a technique in cryptography that has successfully been applied to various ML tasks with structured data (see e.g. [14, 19, 21, 40] and references therein).

To the best of our knowledge there are no existing DP or SMC based solutions for PP feature extraction and classification of unstructured texts. Defenses against authorship attribution attacks that fulfill DP in text classification have been proposed [53]. These methods rely on distortion of term frequency vectors and result in loss of accuracy. In this paper we address a different challenge: we assume that Bob knows Alice, so no authorship obfuscation is needed. Instead, we want to process Alice's text with Bob's classifier, without Bob learning what Alice wrote, and without accuracy loss. To the best of our knowledge, Costantino et al. [15] were the first to propose PP feature extraction from text. In their solution, which is based on homomorphic encryption (HE), Bob learns which of his lexicon's words are present in Alice's tweets, and classification of a single tweet with a model with less than 20 features takes 19 minutes. Our solution does not leak any information about Alice's words to Bob, and classification is done in seconds, even for a model with 500 features.

Below we present existing work that is related to some of the building blocks we use in our PP text classification protocol (see Section 3.1).

*Private equality tests* have been proposed in the literature based on several different flavors [3]. They can be based on Yao Gates, Homomorphic Encryption, and generic SMC [52]. In our case, we have chosen a simple protocol that depends solely on additions and multiplications over a binary field. While different (and possibly more efficient) comparison protocols could be used instead, they would either require additional computational assumptions or present a marginal improvement in performance for the parameters used here.

*Our private feature extraction* can be seen as a particular case of private set intersection (PSI). PSI is the problem of securely computing the intersection of two sets without leaking any information except (possibly) the result, such as identifying the intersection of the set of words in a user's text message with the hate speech lexicon used by the classifier. Several paradigms have been proposed to realize PSI functionality, including a Naive hashing solution, Server-aided PSI, and PSI based on oblivious transfer extension. For a complete survey, we refer to Pinkas et al. [45]. In our protocol for PP text classification, we implement private feature extraction by a straightforward application of our equality test protocol. While more efficient protocols could be obtained by using sophisticated hashing techniques, we have decided to stick with our direct solution since it has no probability of failure and works well for the input sizes needed in our problem. For larger input sizes, a more sophisticated protocol would be a better choice [45].

We use two protocols for the secure classification of feature vectors: an existing protocol $\pi_{\mathsf{LR}}$ for *secure classification with LR models* [19]; and a novel *secure AdaBoost classification protocol*. The logistic regression protocol uses solely additions and multiplications over a finite field. The secure AdaBoost classification protocol is an novel optimized protocol that uses solely decision trees of depth one, binary features and a binary output. All these characteristics were used in order to speed up the resulting protocol. The final secure AdaBoost classification protocol uses only two secure inner products and one secure comparison.

*Generic protocols for private scoring of machine learning models* have been proposed in [8]. The solutions proposed in [8] cannot be used in our setting since they assume that the features' description are publicly known, and thus can be computed locally by Alice and Bob. However, in our case, the features themselves are part of the model and cannot be made public.

Finally, we note that while we implemented our protocols using our own framework for privacy-preserving machine learning [8], any other generic framework for SMC could be also used in principle [47, 22, 41].

## 6   Conclusion

In this paper we have presented the first provably secure method for privacy-preserving (PP) classification of unstructured text. We have provided an analysis of the correctness and security of solution. As a side result, we also present a novel protocol for binary classification over binary input features with an ensemble of decisions stumps. An implementation of the protocols in Java, run on AWS machines, allowed us to classify text messages securely within seconds. It is important to note that this run time (1) includes both secure feature extraction and secure classification of the extracted feature vector; (2) includes both computation and communication costs, as the parties involved in the protocol were run on separate machines; (3) is two orders of magnitude better than the only other existing solution, which is based on HE. Our results show that in order to make PP text classification practical, one needs to pay close attention not only to the underlying cryptographic protocols but also to the underlying ML algorithms. ML algorithms that would be a clear choice when used in the clear might not be useful at all when transferred to the SMC domain. One has to optimize these ML algorithms having in mind their use within SMC protocols. Our results provide the first evidence that provably secure PP text classification is feasible in practice.

## Footnotes

*Guest Professor at Dept. of Applied Mathematics, Computer Science, and Statistics, Ghent University

[2]https://www.bark.us/, https://kidbridge.com/, https://www.webwatcher.com/

[3]https://bitbucket.org/uwtppml

[4]This technique for secure multiplication was originally proposed by Beaver [7] and is regularly used to enable very efficient solutions both in the context of PPML [20, 17, 33, 19] as well as in other applications, e.g., [48, 28, 27, 38, 50, 18].

[5]The hash function is defined as $((a \cdot N + b) \mod p) \mod 2^l - 1$ where $p$ is a prime and $a$ and $b$ are random numbers less than $p$. In our experiments, $p = 1,301,081$, $a = 972$, and $b = 52,097$.

[6]In our case the result of the classification is disclosed to Bob (the party that owns the model) instead of Alice (who has the original input to be classified) as in [19], however it is trivial to modify their protocol so that the final secret share is open towards Bob instead of Alice. Note also that in our case, the feature vector that is used for the classification is already secret shared between Alice and Bob, while in their protocol Alice holds the feature vector, which is then secret shared in the first step of the protocol. This modification is also trivial and does not affect the security of the protocol.

[7]https://bitbucket.org/uwtppml

[8]https://bitbucket.org/uwtppml

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
