[Supplementary Material]

# Appendix A    Correctness and Security Analysis of Protocols

## A.1    Security Model

The gold standard model for proving the security of cryptographic protocols nowadays is the Universal Composability (UC) framework [9] and it is the security model that we use in this work. Protocols that are proven UC-secure enjoy strong securities guarantees and can be arbitrary composed without compromising the security. In short, it is the most adequate model to use when the protocols need to be executed in complex environments such as the Internet, and it additionally allows a modular design of bigger protocols. In this work protocols with two parties, Alice and Bob, are considered and in the following we present an overview of the UC framework for this setting. We refer interested readers to the book of Cramer et al. [16] for more details and the most general definitions.

Apart from the protocol participants, Alice and Bob, there are also an adversary $\mathcal{A}$, an ideal world adversary $\mathcal{S}$ (also known as the simulator) and an environment $\mathcal{Z}$ (which captures everything that happens outside of the instance of the protocol that is being analyzed, and therefore is the one giving the inputs and getting the outputs from the protocol). All these entities are assumed to be interactive Turing machines. The network is assumed to be under adversarial control and therefore $\mathcal{A}$ is the one that delivers the messages between Alice and Bob. In addition to controlling the network scheduling, $\mathcal{A}$ can also corrupt Alice or Bob, in which case he gains the total control over the corrupted party and learn its complete state. For defining the security of the protocol, an ideal functionality $\mathcal{F}$ is defined, which captures the idealized version of what the protocol is supposed to achieve and communicates directly with Alice and Bob to receive the inputs and delivering the outputs of the protocol (in the ideal world, that is all that Alice and Bob do). Then to prove the security of the protocol $\pi$, we show that for every possible adversary $\mathcal{A}$ there exists a simulator $\mathcal{S}$ such that no environment $\mathcal{Z}$ can distinguish between a real world execution with Alice, Bob and the adversary $\mathcal{A}$ running the protocol $\pi$ and the ideal world execution with the ideal functionality $\mathcal{F}$, the simulator $\mathcal{S}$ and the dummy version of Alice and Bob that just forward the inputs and outputs between $\mathcal{F}$ and $\mathcal{S}$. Formally:

**Definition A.1 ([9])** *A protocol $\pi$ UC-realizes an ideal functionality $\mathcal{F}$ if, for every possible adversary $\mathcal{A}$, there exists a simulator $\mathcal{S}$ such that, for every possible environment $\mathcal{Z}$, the view of the environment $\mathcal{Z}$ in the real world execution with $\mathcal{A}$, Alice and Bob executing the protocol $\pi$ (with security parameter $\lambda$) is computationally indistinguishable from the view of $\mathcal{Z}$ in the ideal world execution with the functionality $\mathcal{F}$, the simulator $\mathcal{S}$ and the dummy Alice and Bob, where the probability distribution is taken over the randomness used by all entities.*

**Adversarial Model:** We consider honest-but-curious adversaries. Honest-but-curious adversaries follow the protocol instructions correctly, but try to learn additional information. We only consider static adversaries, for which the set of corrupted parties is chosen before the start of the protocol execution and does not change. A version of the UC theorem for the case of honest-but-curious adversaries is given in Theorem 4.20 of Cramer et al. [16].

**Setup Assumption:** It is a well-known fact that secure two-party computation (and also secure multi-party computation) can only achieve UC-security using a setup assumption [10, 11]. Multiple setup assumptions were used previously to achieve UC-security for secure computation protocols, including: the availability of a common reference string [10, 11, 43], the availability of a public-key infrastructure [4], the random oracle model [36, 5], the existence of noisy channels between the parties [25, 29], and the availability of signature cards [37] or tamper-proof hardware [39, 23, 26]. In this work the commodity-based model [7] is used as the setup assumption. In this model there exists a trusted initializer that pre-distributed correlated randomness to Alice and Bob during a setup phase. This setup phase is run before the protocol execution (and in fact can be performed even before Alice and Bob get to know their inputs), and the trusted initializer does not participate in any other point of the protocol. The commodity-based model was used in many previous works, e.g., [48, 28, 27, 38, 50, 20, 17, 18, 33, 19]. The trusted initializer is modeled by the ideal functionality $\mathcal{F}_{\mathsf{TI}}^{\mathcal{D}}$ described in Figure 3.

**Simplifications:** The simulation strategy in our proofs is in fact very simple: all the computations are performed using secret sharings and all the protocol messages look uniformly random from the point of view of the receiver, with the single exception of the openings of the secret sharings. Nevertheless, the messages that open a secret sharing can be straightforwardly simulated using the outputs of the respective functionalities. In the ideal world, the simulator $\mathcal{S}$ has the leverage of being

---

**Functionality $\mathcal{F}_{\mathsf{TI}}^{\mathcal{D}}$**

$\mathcal{F}_{\mathsf{TI}}^{\mathcal{D}}$ is parametrized by an algorithm $\mathcal{D}$. Upon initialization run $(D_A, D_B) \overset{\$}{\leftarrow} \mathcal{D}$ and deliver $D_A$ to Alice and $D_B$ to Bob.

---

Figure 3: The Trusted Initializer Functionality.

---

**Functionality $\mathcal{F}_{\mathsf{DMM}}$**

$\mathcal{F}_{\mathsf{DMM}}$ is executed with Alice and Bob is parametrized by the size $q$ of the ring and the dimensions $(i, j)$ and $(j, k)$ of the matrices.

**Input:** Upon receiving a message from Alice/Bob with her/his shares of $[\![X]\!]_q$ and $[\![Y]\!]_q$, verify if the share of $X$ is in $\mathbb{Z}_q^{i \times j}$ and the share of $Y$ is in $\mathbb{Z}_q^{j \times k}$. If it is not, abort. Otherwise, record the shares, ignore any subsequent message from that party and inform the other party about the receipt.

**Output:** Upon receipt of the inputs from both Alice and Bob, reconstruct $X$ and $Y$ from the shares, compute $Z = XY$ and create a secret sharing $[\![Z]\!]_q$. Before the deliver of the output shares, a corrupt party fix its share of the output to any constant value. In both cases the shares of the uncorrupted parties are then created by picking uniformly random values subject to the correctness constraint.

---

Figure 4: The Distributed Matrix Multiplication Functionality.

the one responsible for simulating all the ideal functionalities other than the one whose security is being analyzed (including the trusted initializer functionality $\mathcal{F}_{\mathsf{TI}}^{\mathcal{D}}$), and he can easily use this fact to perform a perfect simulation. For this reason the real and ideal world are indistinguishable for any environment $\mathcal{Z}$ and achieve perfect security.

The messages of the ideal functionalities are formally public delayed outputs, i.e., first the simulator is asked whether it allows the message to be delivered (this is due to the fact that in the real world the adversary controls the scheduling of the network), and the message is only delivered when $\mathcal{S}$ agrees. And formally, every instance has a session identification. We omit those information from descriptions for the sake of readability.

**Security of the Building Blocks:** The protocol for secure distributed matrix multiplication $\pi_{\mathsf{DMM}}$ UC-realizes the distributed matrix multiplication functionality $\mathcal{F}_{\mathsf{DMM}}$ described in Figure 4 [24, 19]. The protocol for secure comparison $\pi_{\mathsf{DC}}$ UC-realizes the functionality $\mathcal{F}_{\mathsf{DC}}$ described in Figure 5 [34, 19]. The protocol for secure bit-decomposition $\pi_{\mathsf{decomp}}$ UC-realizes the functionality $\mathcal{F}_{\mathsf{decomp}}$ described in Figure 6 [19]. The LR classification protocol $\pi_{\mathsf{LR}}$ UC-realizes the functionality $\mathcal{F}_{\mathsf{LR}}$ described in Figure 7 [19].

The correctness of the equality test protocol $\pi_{\mathsf{EQ}}$ follows from the fact that in the case that $x = y$, then all $r_i$'s will be equal to 1 and therefore $z = \prod_i r_i$ will also be 1. If $x \neq y$, then for at least one value $i$, we have that $r_i = 0$, and therefore $z = 0$. For the simulation, $\mathcal{S}$ executes an internal copy of $\mathcal{A}$ interacting with an instance of $\pi_{\mathsf{EQ}}$ in which the uncorrupted parties use dummy inputs. Note that all the messages that $\mathcal{A}$ receives look uniformly random to him. Since the share multiplication protocol is substituted by $\mathcal{F}_{\mathsf{DMM}}$ using the UC composition theorem, and $\mathcal{S}$ is the one responsible for simulating $\mathcal{F}_{\mathsf{DMM}}$ in the ideal world, $\mathcal{S}$ can leverage this fact in order to extract the share that any corrupted party have of the value $x_i + y_i$, let the extracted value of the corrupted party be denoted by $v_{i,C}$. The simulator then pick random values $x_{i,C}, y_{i,C} \in \{0, 1\}$ such that $x_{i,C} + y_{i,C} = v_{i,C} \mod 2$ and submit these values to $\mathcal{F}_{\mathsf{EQ}}$ as being the shares of the corrupted party for $x_i$ and $y_i$ (note that the result of $\mathcal{F}_{\mathsf{EQ}}$ only depends on the values of $x_i + y_i \mod 2$). $\mathcal{S}$ is also able to fix the output share of the corrupted party in $\mathcal{F}_{\mathsf{EQ}}$ so that it matches the one in the instance of $\pi_{\mathsf{EQ}}$. This is a perfect

---

**Functionality $\mathcal{F}_{\mathsf{DC}}$**

$\mathcal{F}_{\mathsf{DC}}$ is parametrized by the bit-length $\ell$ of the values being compared.

**Input:** Upon receiving a message from Alice/Bob with her/his shares of $[\![x_i]\!]_2$ and $[\![y_i]\!]_2$ for all $i \in \{1, \ldots, \ell\}$, record the shares, ignore any subsequent messages from that party and inform the other party about the receipt.

**Output:** Upon receipt of the inputs from both parties, reconstruct $x$ and $y$ from the bitwise shares. If $x \geq y$, then create and distribute to Alice and Bob the secret sharing $[\![1]\!]_2$; otherwise the secret sharing $[\![0]\!]_2$. Before the deliver of the output shares, a corrupt party fix its share of the output to any constant value. In both cases the shares of the uncorrupted parties are then created by picking uniformly random values subject to the correctness constraint.

---

Figure 5: The Distributed Comparison Functionality.

---

**Functionality $\mathcal{F}_{\mathsf{decomp}}$**

$\mathcal{F}_{\mathsf{decomp}}$ is parametrized by the bit-length $\ell$ of the value $x$ being converted from an additive secret sharing $[\![x]\!]_q$ in $\mathbb{Z}_q$ to additive bitwise secret sharings $[\![x_i]\!]_2$ in $\mathbb{Z}_2$ such that $x = x_\ell \cdots x_1$.

**Input:** Upon receiving a message from Alice or Bob with her/his share of $[\![x]\!]_q$, record the share, ignore any subsequent messages from that party and inform the other party about the receipt.

**Output:** Upon receipt of both shares, reconstruct $x$, compute its bitwise representation $x_\ell \cdots x_1$, and for $i \in \{1, \ldots, \ell\}$ distribute new secret sharings $[\![x_i]\!]_2$ of the bit $x_i$. Before the output deliver, the corrupt party fix its shares of the outputs to any constant values. The shares of the uncorrupted parties are then created by picking uniformly random values subject to the correctness constraints.

---

Figure 6: The Bit-Decomposition Functionality.

---

**Functionality $\mathcal{F}_{\mathsf{LR}}$**

$\mathcal{F}_{\mathsf{LR}}$ computes the classification according to a logistic regression model with the threshold value set to 0.5. The input feature vector $x$ is secret shared between Alice and Bob.

**Input:** Upon receiving the weight vector $w$, the intercept value $b$ and his shares $[\![x_i]\!]_q$ of the elements of $x$ from Bob, or her shares $[\![x_i]\!]_q$ of the elements of $x$ from Alice, store the information, ignore any subsequent message from that party, and inform the other party about the receipt.

**Output:** Upon getting the inputs from both parties, reconstruct the feature vector $x$, compute the value $\mathsf{sign}\,(\langle x, w \rangle + b)$ and output it to Bob as the class prediction.

---

Figure 7: The Logistic Regression Classification Functionality.

Figure 8: The Equality Test Functionality.

simulation strategy, no environment $\mathcal{Z}$ can distinguish the ideal and real worlds and therefore $\pi_{\mathsf{EQ}}$ UC-realizes $\mathcal{F}_{\mathsf{EQ}}$.

The correctness of the secure feature extraction protocol $\pi_{\mathsf{FE}}$ follows directly from the fact that each $x_{ij}$ is equal to 1 if, and only if, $a_j = b_i$, and therefore $x_i = \sum_j x_{ij}$ is equal to 1 if, and only if, $b_i$ is equal to some element of $A$. In the ideal world, the simulator $\mathcal{S}$ runs internally a copy of $\mathcal{A}$ and an execution of $\pi_{\mathsf{FE}}$ with dummy inputs for the uncorrupted parties. All the messages from the uncorrupted parties look uniformly random from $\mathcal{A}$'s point of view, and therefore the simulation is perfect. $\mathcal{S}$ uses the leverage of being responsible for simulating $\mathcal{F}_{\mathsf{EQ}}$ ($\pi_{\mathsf{EQ}}$ is substituted by $\mathcal{F}_{\mathsf{EQ}}$ using the UC composition theorem) in order to extract the inputs of any corrupted party and forward it to $\mathcal{F}_{\mathsf{FE}}$. No environment $\mathcal{Z}$ can distinguish the ideal world from the real one, and thus $\pi_{\mathsf{FE}}$ UC-realizes $\mathcal{F}_{\mathsf{FE}}$.

In the case of the conversion protocol $\pi_{\mathsf{2toQ}}$ the correctness of the protocol execution follows straightforwardly: since $x = x_a + x_B \mod 2$, then $z = x_A + x_B - 2x_Ax_B$ is such that $z = x$ for all possible values $x_A, x_B \in \{0, 1\}$. As for the security, the simulator $\mathcal{S}$ runs internally a copy of the adversary $\mathcal{A}$ and simulates to him an execution of the protocol $\pi_{\mathsf{2toQ}}$ using dummy inputs for the uncorrupted parties. As all the messages from the uncorrupted parties look uniformly random from the adversary point of view, and so the simulation is perfect. The simulator can use the fact that it is the one simulating the multiplication functionality $\mathcal{F}_{\mathsf{DMM}}$ (the secret sharing multiplication is substituted by $\mathcal{F}_{\mathsf{DMM}}$ using the UC composition theorem) in order to extract the share of any corrupted party and fix the input to/output from $\mathcal{F}_{\mathsf{2toQ}}$ appropriately, so that no environment $\mathcal{Z}$ can distinguish the real and ideal worlds. Hence $\pi_{\mathsf{2toQ}}$ UC-realizes $\mathcal{F}_{\mathsf{2toQ}}$.

The AdaBoost classification protocol $\pi_{\mathsf{AB}}$ is trivially correct for the case of binary features and output class, and decision stumps. In the simulation, $\mathcal{S}$ runs an internal copy of $\mathcal{A}$ interacting with a simulated instance of $\pi_{\mathsf{AB}}$ that uses dummy inputs for the uncorrupted parties. $\pi_{\mathsf{IP}}$ is substituted by $\mathcal{F}_{\mathsf{DMM}}$ using the UC composition theorem. $\mathcal{S}$ uses the leverage of simulating $\mathcal{F}_{\mathsf{DMM}}$ in order to extract the shares of the feature vector belonging to a corrupted party, as well as the weighted probability vectors $y$ and $z$ if Bob is corrupted. $\mathcal{S}$ can then give these extracted inputs to $\mathcal{F}_{\mathsf{AB}}$. No environment can distinguish the real and ideal worlds since the simulation is perfect, and thus $\pi_{\mathsf{AB}}$ UC-realizes $\mathcal{F}_{\mathsf{AB}}$.

**Security of the Privacy-Preserving Text Classification Solutions:**

The protocol $\pi_{\mathsf{TC-LR}}$ simply executes sequentially the protocols $\pi_{\mathsf{FE}}$, $\pi_{\mathsf{2toQ}}$ and $\pi_{\mathsf{LR}}$. Given that these protocols UC-realize $\mathcal{F}_{\mathsf{FE}}$, $\mathcal{F}_{\mathsf{2toQ}}$ and $\mathcal{F}_{\mathsf{LR}}$, respectively, they can be substituted by the functionalities using the UC composition theorem. Note that the sequential composition of those functionalities trivially perform the same computation as $\mathcal{F}_{\mathsf{TC-LR}}$, and no information other than the output of the classification is revealed (all the intermediate values are kept as secret sharings). In the ideal world $\mathcal{S}$ simulates an internal copy of the adversary $\mathcal{A}$ running $\pi_{\mathsf{TC-LR}}$ and using dummy inputs for the uncorrupted parties. The simulator $\mathcal{S}$ can easily extract all the information (from the corrupted parties) that it needs to provide to $\mathcal{F}_{\mathsf{TC-LR}}$ by using the leverage of being responsible for simulating

**Functionality $\mathcal{F}_{\mathsf{FE}}$**

$\mathcal{F}_{\mathsf{FE}}$ is parametrized by the sizes $m$ of Alice's set and $n$ of Bob's set, and the bit-length $\ell$ of the elements.

**Input:** Upon receiving a message from Alice with her set $A = \{a_1, a_2, \ldots, a_m\}$ or from Bob with his set $B = \{b_1, b_2, \ldots, b_n\}$, record the set, ignore any subsequent messages from that party and inform the other party about the receipt.

**Output:** Upon receipt of the inputs from both parties, define the binary feature vector $x$ of length $n$ by setting each element $x_i$ to 1 if $b_i \in A$, and to 0 otherwise. Then create and distribute to Alice and Bob the secret sharings $[\![x_i]\!]_2$. Before the deliver of the output shares, a corrupt party fix its share of the output to any constant value. In both cases the shares of the uncorrupted parties are then created by picking uniformly random values subject to the correctness constraint.

Figure 9: The Secure Feature Extraction Functionality.

**Functionality $\mathcal{F}_{\mathsf{2toQ}}$**

$\mathcal{F}_{\mathsf{2toQ}}$ is parametrized by the size of the field $q$.

**Input:** Upon receiving a message from Alice/Bob with her/his share of $[\![x]\!]_2$, record the share, ignore any subsequent messages from that party and inform the other party about the receipt.

**Output:** Upon receipt of the inputs from both parties, reconstruct $x$, then create and distribute to Alice and Bob the secret sharing $[\![x]\!]_q$. Before the deliver of the output shares, a corrupt party fix its share of the output to any constant value. In both cases the shares of the uncorrupted parties are then created by picking uniformly random values subject to the correctness constraint.

Figure 10: The Secret Sharing Conversion Functionality.

**Functionality $\mathcal{F}_{\mathsf{AB}}$**

$\mathcal{F}_{\mathsf{AB}}$ computes the classification according to AdaBoost with multiple decision stumps. All the features are binary and the output class is also binary. The input feature vector $x$ is secret shared between Alice and Bob. The model specified by Bob can be expressed in a simplified way by two weighted probability vectors $y = (y_{1,0}, y_{1,1}, \ldots, y_{n,0}, y_{n,1})$ and $z = (z_{1,0}, z_{1,1}, \ldots, z_{n,0}, z_{n,1})$. For the $i$-th decision stump: $y_{i,k}$ is the weighted probability (i.e., a probability multiplied by the weight of the $i$-th decision stump) that the model assigns to the output class being 0 if $x_i = k$, and $z_{i,k}$ is defined similarly for the output class 1.

**Input:** Upon receiving the vectors $y$ and $z$ and his shares $[\![x_i]\!]_q$ of the elements of the feature vector $x$ from Bob, or her shares $[\![x_i]\!]_q$ of the elements of $x$ from Alice, store the information, ignore any subsequent message from that party, and inform the other party about the receipt.

**Output:** Upon getting the inputs from both parties, reconstruct the feature vector $x$ and let $w = (1 - x_1, x_1, 1 - x_2, x_2, \ldots, 1 - x_n, x_n)$. If $\langle w, z \rangle \geq \langle w, y \rangle$, output the class prediction 1 to Bob; otherwise output 0.

Figure 11: The AdaBoost Classification Functionality.

---

**Functionality** $\mathcal{F}_{\mathsf{TC-LR}}$

$\mathcal{F}_{\mathsf{TC-LR}}$ computes the privacy-preserving text classification according to a logistic regression model with the threshold value set to 0.5. It is parametrized by the sizes $m$ of Alice's set and $n$ of Bob's set, and the bit-length $\ell$ of the elements.

**Input:** Upon receiving a message from Alice with her set $A = \{a_1, a_2, \ldots, a_m\}$ or from Bob with his set $B = \{b_1, b_2, \ldots, b_n\}$, the weight vector $w$ and the intercept value $b$, record the values, ignore any subsequent messages from that party and inform the other party about the receipt.

**Output:** Upon getting the inputs from both parties, define the feature vector $x$ of length $n$ as follows: $x_i = 1$ if $b_i \in A$; and $x_i = 0$ otherwise. Compute the value $\mathsf{sign}\left(\langle x, w \rangle + b\right)$ and output it to Bob as the class prediction.

---

Figure 12: The Functionality for Privacy-Preserving Text Classification with Logistic Regression.

---

**Functionality** $\mathcal{F}_{\mathsf{TC-AB}}$

$\mathcal{F}_{\mathsf{TC-AB}}$ computes the privacy-preserving text classification according to AdaBoost with multiple decision stumps. It is parametrized by the sizes $m$ of Alice's set and $n$ of Bob's set, and the bit-length $\ell$ of the elements. All the features are binary and the output class is also binary. The model specified by Bob can be expressed in a simplified way by two weighted probability vectors $y = (y_{1,0}, y_{1,1}, \ldots, y_{n,0}, y_{n,1})$ and $z = (z_{1,0}, z_{1,1}, \ldots, z_{n,0}, z_{n,1})$. For the $i$-th decision stump: $y_{i,k}$ is the weighted probability (i.e., a probability multiplied by the weight of the $i$-th decision stump) that the model assigns to the output class being 0 if the feature $x_i = k$, and $z_{i,k}$ is defined similarly for the output class 1.

**Input:** Upon receiving a message from Alice with her set $A = \{a_1, a_2, \ldots, a_m\}$ or from Bob with his set $B = \{b_1, b_2, \ldots, b_n\}$, $y$ and $z$, record the values, ignore any subsequent messages from that party and inform the other party about the receipt.

**Output:** Upon getting the inputs from both parties, define the feature vector $x$ of length $n$ as follows: $x_i = 1$ if $b_i \in A$; and $x_i = 0$ otherwise. Let $w = (1-x_1, x_1, 1-x_2, x_2, \ldots, 1-x_n, x_n)$. If $\langle w, z \rangle \geq \langle w, y \rangle$, output the class prediction 1 to Bob; otherwise output 0.

---

Figure 13: The Functionality for Privacy-Preserving Text Classification with Adaboost.

$\mathcal{F}_{\mathsf{FE}}$, $\mathcal{F}_{\mathsf{2toQ}}$ and $\mathcal{F}_{\mathsf{LR}}$ in the ideal world. Therefore no environment $\mathcal{Z}$ can distinguish the real world from the ideal world, and $\pi_{\mathsf{TC-LR}}$ UC-realizes $\mathcal{F}_{\mathsf{TC-LR}}$.

Similarly, the protocol $\pi_{\mathsf{TC-AB}}$ just runs sequentially the protocols $\pi_{\mathsf{FE}}$, $\pi_{\mathsf{2toQ}}$ and $\pi_{\mathsf{AB}}$, that can be substituted by $\mathcal{F}_{\mathsf{FE}}$, $\mathcal{F}_{\mathsf{2toQ}}$ and $\mathcal{F}_{\mathsf{AB}}$ using the UC composition theorem. The result of the computation is trivially the same as in $\mathcal{F}_{\mathsf{TC-AB}}$, and no additional information is revealed. $\mathcal{S}$ runs internally a copy of $\mathcal{A}$ interacting with a simulated instance of $\pi_{\mathsf{TC-AB}}$ (using dummy inputs for the uncorrupted parties) and can easily extract from the corrupted parties all the information that it must provide to $\mathcal{F}_{\mathsf{TC-AB}}$ by using the leverage of being responsible for simulating $\mathcal{F}_{\mathsf{FE}}$, $\mathcal{F}_{\mathsf{2toQ}}$ and $\mathcal{F}_{\mathsf{AB}}$ in the ideal world. No environment $\mathcal{Z}$ can distinguish the real and ideal worlds, and therefore $\pi_{\mathsf{TC-AB}}$ UC-realizes $\mathcal{F}_{\mathsf{TC-AB}}$.