[Reviews · NeurIPS 2019]

Reviewer 1



This paper studies the problem of privacy-preserving text classification. More specifically, the authors develop a SMC based method for logistic regression and AdaBoost models. Experiments on a real dataset validate the performance of the proposed method. However, the current paper has some issues need to be addressed: 1.The proposed method can only protect the privacy at the inference procedure. To achieve this goal, some other privacy-preserving frameworks, such as differential privacy (DP), can also be used. For example, DP can directly add some small noise to protect the privacy at the inference procedure, which is very simple and efficient. In this sense, the contribution of the current paper is not very clear. 2.The proposed method can only be applied to some very specific models, like uni/bi gram logistic regression and AdaBoost models, which are not the state-of-the art models in both theory and practice for text classification. As a result, the proposed method may not have too much impact. 3.In experiments, there is no baseline result, i.e., the result for non-private method. And the results in terms of accuracy and time seem to be not good. --- After reading the author response, I agree that the proposed SMC framework is a contribution to the secure ML community, and I would like to increase my score.

Reviewer 2



The secret sharing method has been used a lot in multi-party computation. Yucel Saygin wrote several papers on secure clustering using secret sharing. In contrast, the authors force their computations to remain on a ring and could thus map to the 2 adic ring. The mapping and equations are straightforward and not surprising. But the contribution is nice and could stimulate future research. Writing is sound and clear.

Reviewer 3



The authors present a privacy-preserving protocol for learning text classifiers on short texts using secure multiparty communication (SMC). Unlike differential privacy under the central model, a more popular framework at the moment for making it difficult to distinguish the presence or absence of individuals in training data for a model, this protocol aims to ensure that a pretrained classifier may be used on new text data without leaking that data to the classifier's owner. Though the underlying classifier is not a SOTA solution to the test classification problem, hate speech detection, it is a nontrivial classifier of text and can classify a single example in a matter of seconds, substantially improving over the performance of approaches using homomorphic encryption. The authors test their approach on a collection of 10,000 tweets with binary labels describing whether they are hate speech, demonstrating the effectiveness of this tool in aiding automatic moderation of sensitive content. I want to be open that I am not an expert on SMC, and my primary knowledge of privacy-preserving ML is through differential privacy and natural language processing. However, I was impressed at the authors' effective tradeoff of the detail necessary to present their technical contribution in terms of secret-sharing protocols and corresponding proofs of correctness and clarity in their explanation of the role of SMC. While the end result is, at face value, a relatively specific classification algorithm (AdaBoost for short text applied to small hate speech), the building blocks presented here should provide some inspiration for more substantial efforts of this kind, and I'm excited to see where that leads. My biggest questions about the technical contribution of this paper related to the feature representations and transmission of information related to feature vectors, but I think I've mostly worked them out, and I just wanted to state them explicitly here to make sure I understand correctly: - The fact that feature vectors may be redundant (e.g., knowing which bigrams are present uniquely determines which unigrams are present if features have not been pruned) does not affect SMC privacy guarantees. I believe this guarantee comes from the way polynomials for secret sharing are assembled with respect to the total possible domain of values available for a vector. - While the total feature set available in the text collection is much larger than would typically be considered in a text classification setting, at any time only the lexicon of an individual tweet + the lexicon of the features used in Bob's classifer will be used in the lexicon merging phase, so $\pi_{FE}$ will only be considering hundreds of features at a time, not 123,482. This seems to be confirmed in Section 4.1. The chief obstacle to using this sort of method more regularly is that, while vastly faster than HE, several seconds per single classification with a simple classifier is still prohibitively slow for lots of industry applications, and given the size of machines used, it seems like scaling up the responsive servers for this sort of application might not solve the problem. One thing I wondered about was the specific operations that were most time- and memory-consumptive in applying a classifier...is it possible to get a sense of the timing information for the individual operations instead of full classifications? Would information about the amount of time taken to merge the lexicons to produce the feature vector be susceptible to a timing attack? A less important but still valuable gap this paper should consider filling is with regards to claims about relevant DP work in the field. While I concur that most work has focused on hiding training instances, not securely transmitting novel classification data, I think work towards locally-private online learning with DP done by e.g. Apple and Google is still relevant to this domain and bears slightly more development as a discussion topic with which to draw contrast. Smaller notes not affecting my score: - Some attention to phrasing in the first paragraph might be good: given that invasion of childrens' privacy through moderation isn't a totally uncontentious topic when e.g. they're dealing with toxic home situations or abusive relationships, it might not be good to frame it as a "clear benefit" - I think the citation in line 45 might be intended to be [14], not [33]? - The fact that a hashing technique could apply here is actually extremely promising, given hashed text tends to still work well for classifiers and, if features were hashed in the learned model, the small amount of noise added may even help regularize it slightly. It might be worth selling this more. Overall, I was really excited to read this paper, as it's polished work, and am hopeful for its future! -- Based on the author response, I am leaving my review as-is; I am grateful to the authors' address of the high-level points about DP in interaction with SMC but definitely want to make sure to keep their attention on the pieces above to clarify/explain more.

[Author Response · NeurIPS 2019]

**Reviewer 1** - **(1).** The reviewer's comments show a misunderstanding concerning what is achieved by our protocol and what is achievable by differential privacy (DP). In our model, a data holder wants to score an input $x_i$ against a model $M$ held by another party (model holder) such that, at the end of the protocol, no information about the input is leaked to the model holder (beyond the result of the classification) and no information about the model should leak to the data holder. Information about the model and the data also *should not be available to any other party involved in the computation*. Differential privacy is not useful in this scenario. In the central DP set-up, the data collector accesses the entire data set. Upon receiving a question, the data collector computes an answer based on the data set, adds noise to the answer and sends it to the party asking the question. In this case, unlike with the SMC approach, there is loss of accuracy because of the noise, and more importantly, all information, including the question, the dataset and the answer, is leaked to the data collector. In the local DP set-up, data owners add noise to their data entries and send them to a third party (the model holder / data collector). The data collector uses the noisy entries ($X = x_1, x_2, ...x_n$) to answer a question $A(X)$, $x_i$ represents the local data plus noise. The overall system is said to be differentially private if the view of the data collector does not change much if the data set is modified in just one entry. While in this scenario, privacy for the local inputs is possible, *the question $A$ cannot depend solely on a single entry* by the very definition of differential privacy. DP cannot be used to single out individual "bad" entries. DP deals with global characteristics of the data set $X$. Our solution, on the other hand receives a single entry $x_i$ and outputs the classification of $x_i$ such that the only information leaked to the model holder is the class label and no information is leaked to the data holder at all, while having no loss in accuracy and without any trusted entity receiving information about the model or the input data. It is misleading to directly compare DP with SMC. They are different tools for achieving different notions of privacy in different situations. In many ways, they complement each other.

**(2).** It is an intrinsic characteristic of SMC that the function to be computed privately has to be represented as a circuit of addition and multiplication gates. Therefore, one has to come up with specific circuits and optimizations for each ML technique and algorithm. We have provided a fairly general solution to an important problem: text classification. Developing SMC protocols for all SOTA ML models is far beyond the scope of a single paper. Our work is the very first SMC based method for text classification. While we recognize that deep learning is SOTA for many NLP tasks, LR models on word n-grams for hate speech detection have been observed to be at par with CNN and LSTM model architectures (cfr. [33] Gröndahl et al., AISec 2018). Furthermore, we fully agree with reviewer 3 that many of the building blocks presented in the paper can be re-used or inspire the development of similar SMC building blocks for other kinds of ML models, stimulating future research as also pointed out by reviewer 2. We remark here that we have proved (in the appendix) that our building blocks can be securely composed.

**(3).** We disagree with the reviewer that the results in terms of accuracy and time are not good. The accuracy with SMC is the same as the accuracy in the clear (i.e. for the "non-private method"), in other words there is no loss of accuracy. The classification time is two orders of magnitude better than that of the only existing solution for privacy-preserving text classification and it comes with rigorous proofs of security. As pointed out, fast DP based solutions are of no help here, since the result of the classification depends solely on a single entry of the data set and the input data and the model should remain private.

**Reviewer 2** - We thank reviewer 2 for his/her insightful comments. Indeed, to the best of our knowledge, we have designed the first efficient provably secure protocols for doing private text classification of individual entries. Making sure that all the computations happen over a ring, rather than a field, helped us to reduce the round complexity of our solution. We will cite works of SMC applied to other areas of ML, including clustering.

**Reviewer 3** - We thank reviewer 3 for his/her careful review of our work. The reviewer's understanding of our paper is correct. The reviewer is correct in pointing out that the running time of the proposed protocols (seconds) is still higher than in the clear (a few milliseconds). SMC implementations are still substantially slower than solutions in the clear. Moreover, from an implementation perspective, we have paid a price in making our implementation modular and in Java. We probably could have decreased the running time by implementing everything in assembly and C. An association of Yao Garbled Circuits and Secret Sharing SMC combined with optimal ML protocols could give us an improvement over what we present here (but is beyond the scope of the paper). Improving those running times will demand substantial work from the ML and Cryptography communities, preferably in association. As it is, the biggest bottleneck in our solution is the private feature extraction. Coming up with improvements for it would have a big impact in our protocols. The relation between DP and our work is an interesting one. Locally-private DP offers the possibility of obtaining statistics and ML models trained over a data set distributed over many owners in a private way. It comes at a cost: there is no way to classify individual entries (see answer to Reviewer 1) and there is a substantial loss of accuracy, particularly when the number of entries is not large. However, it is fast. SMC gives us the possibility of scoring single entries against a ML model privately without any loss in accuracy. However, it is slow. We plan on investigating a mix of these two techniques, where SMC is used to decrease the noise/loss in the locally-private ML model. We thank the reviewer for pointing out this discussion topic and we plan on adding it to the paper. We believe that SMC and DP are complementary solutions; bringing the promise of privacy-preserving ML to practice will require an association of these and other paradigms. Exposing the ML community to works like ours is a necessary step towards that direction.

[Meta-Review · NeurIPS 2019]

This paper studies the problem of privacy preserving inference where two parties, one of which is holding a model and the other holding a piece of text, would like to score the text by the model without exchanging neither the text or the model. The authors use secure multi party computation techniques to present a solution to this problem. The methods used in this work are not new. However, building a complete solution of the sort discussed here requires assembling together multiple pieces where for each piece there are multiple solutions to choose from. The authors to a good job at describing the pieces and the reasoning behind every choice they make such that the overall solution will perform well – I see that as a significant contribution. They also present a formal analysis of the security of the model and an empirical evaluation which makes this a well-rounded paper.